# High-Spatial-Resolution OFDR Distributed Temperature Sensor Based on Step-by-Step and Image Wavelet Denoising Methods

**DOI:** 10.3390/s22249972

**Published:** 2022-12-17

**Authors:** Cailing Fu, Pengfei Li, Ronglong Sui, Zhenwei Peng, Huajian Zhong, Xiaoyu Yin, Yiping Wang

**Affiliations:** 1Key Laboratory of Optoelectronic Devices and Systems of Ministry of Education and Guangdong Province, College of Physics and Optoelectronic Engineering, Shenzhen University, Shenzhen 518060, China; 2Shenzhen Key Laboratory of Photonic Devices and Sensing Systems for Internet of Things, Guangdong and Hong Kong Joint Research Centre for Optical Fibre Sensors, Shenzhen University, Shenzhen 518060, China; 3College of Electronic Engineering, Nanjing Xiaozhuang University, Nanjing 211171, China

**Keywords:** optical frequency domain reflectometry, distributed optical fiber sensing, image wavelet denoising

## Abstract

A high-spatial-resolution OFDR distributed temperature sensor based on Au-SMF was experimentally demonstrated by using step-by-step and image wavelet denoising methods (IWDM). The measured temperature between 50 and 600 °C could be successfully demodulated by using SM-IWDM at a spatial resolution of 3.2 mm. The temperature sensitivity coefficient of the Au-SMF was 3.18 GHz/°C. The accuracy of the demodulated temperature was approximately 0.24 °C. Such a method has great potential to expand the temperature measurement range, which is very useful for high-temperature applications.

## 1. Introduction

Fiber optic distributed temperature sensing (DTS) has attracted considerable attention in various applications, such as pipeline flux leakage monitoring [1], fire detection systems [2] and cable fault location [3] for the advantages of distributed measurement capability, long sensing range and electromagnetic interference immunity. To date, several methods have been proposed to realize DTS by using the intrinsic scattering effect in optical fiber, i.e., Raman [4,5], Brillouin [6] and Rayleigh scattering [7,8,9,10]. Among them, DTS based on an optical time domain reflectometer (OTDR) and optical frequency domain reflectometer (OFDR) exhibit a high signal-to-noise ratio (SNR) due to the high intensity of the Rayleigh backscattering (RBS), where the temperature distribution can be demodulated by analyzing the intensity or wavelength change of the RBS signal [7,8,9,10]. The DTS based on OTDR is suitable for long-distance optical fiber link monitoring with meter-level low spatial resolution [7,9]. Compared with DTS based on OTDR, the DTS based on OFDR can achieve a high spatial resolution of a millimeter-level by using continuous optical modulation technology and the frequency domain analysis method [8,10]. However, the existence of coherent noise, white noise and environmental noise reduces the SNR of the sensing signal, resulting in the deterioration of the spatial resolution to centimeter-level [11,12,13]. Recently, micro-cavity array based on dense ultra-short (FBG) array [14] and ultra-weak FBG array [15] fabricated by a femtosecond laser direct technique has been employed to improve the temperature-sensing spatial resolution of the DTS based on OFDR. However, the DTS using FBG array is still quasi-distributed, where the multiplexing capacity is largely dependent on the reflectivity and space between adjacent FBGs [14,15]. Moreover, many data post-processing methods, i.e., phase-domain-interpolation resampling [16], differential relative phase [17] and recursive compensation [18], are also proposed to improve the temperature-sensing spatial resolution, which would further increase the computing time. Among these methods, the temperature range, i.e., high temperature, is limited by using the traditional static reference method. To resolve data at high temperature using SMF, various methods based on the adaptive reference method have been demonstrated to demodulate temperature distribution [19,20,21,22]. For example, Sweeney et al. presented an inchworm method, where the reference was varied in a more sophisticated manner based on a quality metric [21]. Subsequently, a senor-by-sensor inchworm method was further proposed by adaptively varying the reference measurement position by position [20]. Then, a maximum spanning tree was also used to optimize the spectral shift by selecting the best reference measurements for each active measurement in a series [21]. 

Therefore, it is urgent to find an economical method to realize high-spatial resolution DTS.

## 2. Experimental Setup and Methods

A typical OFDR experimental setup for distributed temperature sensing (DTS) was built, as shown in Figure 1. The system consists of two arms: auxiliary arm is used to provide an external clock signal to suppress the nonlinear tuning errors of the swept laser, i.e., tunable laser source (TLS); measurement arm is used to detect the RBS information in the Au-coated single model fiber (Au-SMF). The continuous laser output of the TLS is split into two paths by an optical coupler (OC_1_). Ten percent of the output is launched into the auxiliary arm, i.e., a Michelson interferometer consists of two Faraday rotating mirrors (FRMs). The length of the delay fiber is 70 m, and the obtained signal by balanced photodetector (BPD_1_) is employed to provide trigger signal to the data acquisition (DAQ). Ninety percent of the output is launched into the measurement arm and then is divided into two parts by OC_2_. One part of the light is sent to the Au-SMF through CIR_2_; another part is sent to the polarization controller (PC) to adjust the power of the p/s components. Then, the OC_3_ is used to combine two parts. The obtained beat signal is spilt into p and s components through two polarization beam splitters (PBSs), and detected by BPD_2_ and BPD_3_, and then acquired by DAQ. Moreover, two PBSs are also used to mitigate the polarization signal fading of the measurement arm [23]. In the experiment, the TLS is swept from 1545 to 1555 nm with a speed rate of 40 nm/s, indicating that the range of the sweep frequency of the TLS, i.e., ∆F, is 1250 GHz. Thus, the two-point spatial resolution is 0.08 mm. In addition, the temperature-sensing property of the Au-SMF is investigated by placing the fiber end, i.e., 10 cm, in the furnace (Gemini4857A), where the diameter of the furnace cavity is 8 mm. The temperature is increased from 50 to 600 °C with a step of 50 °C, remaining for 40 min at each temperature measurement point.

Generally, a traditional method, i.e., direct method (DM), based on OFDR is used to demodulate the temperature distribution in the furnace, as illustrated in Figure 2a. It means that the initial temperature, i.e., *T*_0_, and measured temperature, i.e., *T_n_*, are defined as the reference (Ref.) and measurement (Mea.) signals, respectively. Then, the temperature distribution along the FUT can be obtained by the cross-correlation spectral shift between Ref. and Mea. signal, where the accuracy of the obtained temperature is dependent on the similarity between them. However, the similarity deteriorates with the increase in the temperature, resulting in false-peaks or multi-peaks. To overcome this problem, another method, i.e., step-by-step method (SM), is proposed and demonstrated, as shown in Figure 2b. Compared with the DM, multiple transition temperatures, i.e., *T*_1_, *T*_2_, …, *T_n_*_−1_, were selected for the SM to demodulate the measured temperature, i.e., *T_n_*. This indicated that the initial temperature, i.e., *T*_0_, was used as the Ref. signal to demodulate the first transition temperature, i.e., *T*_1_, and then the temperature *T*_1_ was used as the Ref. signal to demodulate the second transition temperature, i.e., *T*_2_, and so on to the (*n*−1)th transition temperature, i.e., *T_n_*_−1_. In this way, the measured temperature, i.e., *T_n_*, could be demodulated by taking the temperature of *T_n_*_−1_ as the Ref. signal. The principle of the temperature demodulation was listed as follows. The signal from the AI acquired by the DAQ could be given by:(1)$$\left(t\right)=2\sqrt{R\left(\tau_{z}\right)}\;E_{0}cos\left\{2\pi \left[f_{0}\tau_{z}+f_{b}t-\frac{1}{2}\gamma \tau_{z}^{2}+\varphi \left(t\right)-\varphi \left(t-\tau_{z}\right)\right]\right\}$$
where $\tau_{z}$ is the delay time between the two arms of the auxiliary arm, $R\left(\tau_{Z}\right)$ is the reflectivity with the fiber attenuation at the delay time of $\tau_{z}$, $f_{0}$ and γ are the initial optical frequency and sweep rate of the tunable laser source (TLS), $E_{0}$ is the amplitude of the optical electric field, $\varphi \left(t\right)-\varphi \left(t-\tau_{z}\right)$ is the phase noise term, respectively. In addition, the value of the beat frequency, i.e., $f_{b}$, is proportional to the time delay, i.e., $\tau_{z}$, with a relationship of $f_{b}=\gamma \tau_{z}$. By the cross-correlation between the measurement Rayleigh backscattering (RBS) and the reference RBS, the RBS shifts, i.e., Δf can be obtained. Then, Δf caused by the temperature variation, i.e., ΔT, can be given by:(2)Δf=KΔT

Therefore, the temperature variation along the FUT can be obtained by measuring the spectral shift, i.e., Δf.

To further improve the spatial resolution for high-temperature sensing, an image wavelet denoising method (IWDM) is used in combination with SM to realize the temperature demodulation. As shown in Figure 2c, the detailed process is as follows. Firstly, the temperature sensor, i.e., Au-SMF, is evenly split into multiple parts of the same length, i.e., *R_i_* and *M_i_* (*i* = 1, 2, 3…) for the Ref. signal (*T*_0_, *T*_1_, …., *T_n_*_−1_) and Mea. signal (*T*_1_, *T*_2_, …, *T_n_*) in time domain, respectively, by use of Fast Fourier Transformation (FFT). Secondly, the p/s components of each part in the time domain are converted to the frequency domain by inverse FFT (IFFT). Thirdly, the cross-correlation operation is performed on the Ref. and Mea. signal, i.e., *T*_0_ and *T*_1_, *T*_1_ and *T*_2_, …, *T_n_*_−1_ and *T_n_*, respectively, after the vector sum of p/s components, corresponding to the spectral shift of ∆*f*_1_, ∆*f*_2_, …, ∆*f_n_*, respectively. Here, a two-dimensional (2D) cross-correlation matrix, i.e., *C*(*z*, ∆*f*), consisting of fiber position, i.e., *z*, and spectral shift, i.e., ∆*f*, can be obtained. Meanwhile, the noise signal can be eliminated by adopting a threshold function due to different wavelet coefficients of the noise and sensing signal. In other words, the wavelet coefficients below the threshold are the noise signal and set to zero, while above the threshold is the sensing signal. Note that the threshold is set to 0.0664, i.e., three times the obtained noise standard deviation, i.e., 0.0197. In this way, an optimized two-dimensional (2D) cross-correlation image matrix, i.e., *C*′(*z*, ∆*f*′), can be obtained by using the remaining wavelet coefficients through inverse wavelet transform, where ∆*f*′ is the optimized spectral shift [11]. Therefore, the total optimized spectral shift, i.e., ∆*f*′, can be given by: (3)$$\Delta f^{\prime }=\Delta f_{1}^{\prime }+\ldots \Delta f_{i}^{\prime }+\ldots \Delta f_{n}^{\prime }=k\left(\Delta T_{1}+\ldots \Delta T_{i}+\ldots \Delta T_{n}\right)$$
where *k* is the temperature response coefficient of Au-coated SMF, and $\Delta T_{i}=T_{i}-T_{i-1}$ (*i* = 1,…, *n*). Consequently, the measured temperature, i.e., *T_n_*, of the furnace can be deduced from the optimized spectral shift, i.e., ∆*f*′. Note that the 2D and 1D methods are defined as using and not using IWDM, respectively.

## 3. Results and Discussions

To better demonstrate the effectiveness of the proposed method, the demodulation results of DM combined with the 1D and 2D methods, i.e., DM-1D and DM-2D and SM combined with the 1D and 2D methods, i.e., SM-1D and SM-2D, at the furnace temperature of 600 °C were compared. In the experiment, the FUT between 9.85 and 10.0 m was placed in the furnace. As shown in Figure 3a, the signal between 9.85 and 10.00 m along the FUT was completely submerged in noise by using the DM-1D. Note that in the DM, the Ref. signal and Mea. signal are defined as signals acquired at the temperature of 50 °C and 600 °C, respectively. Compared with DM-1D, the temperature distribution demodulated by DM-2D is slightly improved, but it is still indistinguishable, as illustrated in Figure 3b. This indicates that neither DM-1D nor DM-2D can demodulate the temperature distribution when the temperature is 600 °C.

Subsequently, the SM combined with the 1D and 2D methods, i.e., SM-1D and SM-2D, is also employed to demodulate the measured temperature, i.e., 600 °C. In the experiment, the temperature of the furnace was increased from 50 °C to 600 °C with a step of 50 °C, and maintained for 1 h at each temperature. Moreover, the RBS signal at each measured temperature was obtained by the OFDR illustrated in Figure 1. Note that in the SM, the Ref. signal and Mea. signal are defined as signals acquired at the temperatures of 550 °C and 600 °C, respectively. As shown in Figure 4a, the trend of the demodulated temperature distribution by using DM-1D could be roughly observed, but the fluctuation of the spectral shift was up to 455.3 GHz. Compared with DM-1D, the fluctuation of the spectral shift was obviously improved by using SM-1D. As shown in Figure 4b, the temperature distribution of the heating region between 9.85 and 10.0 m was successfully demodulated, where the spectral shift between the Ref. temperature, i.e., 550 °C, and Mea. temperature, i.e., 600 °C, was 92.2 GHz, i.e., ∆*fn’* = 92.2 GHz. This indicated that the temperature distribution at the temperature of 600 °C could be accurately recovered by using SM-2D. 

Moreover, the shifts induced by the temperature change were calculated by DM-1D, DM-2D, SM-1D and SM-2D, when the temperature was increased from 50 to 600 °C with a step of 50 °C. As shown in Figure 5a,b, the spectral shift could not be distinguished by using DM, which was the same as Figure 3. As shown in Figure 5c, the spectral shift could be clearly identified by using SM-1D when the temperature was lower than 400 °C, and significant fluctuations could be clearly observed when the temperature was higher than 400 °C, where the fluctuations became larger with the increase in temperature. As shown in Figure 5d, the temperature distribution of three areas, i.e., the outside, top and inside of the furnace represented by the pink, orange and cyan boxes, respectively, at each temperature could be well demodulated by using DM-2D. Note that the sensing spatial resolution was 3.2 mm, where the number of data points was 40. An obvious non-uniform temperature profile between 9.65 and 9.90 m, i.e., the outside and top of the furnace cavity, was obtained, while a relatively uniform temperature profile between 9.90 and 10.0 m, i.e., the inside the furnace cavity, was obtained. 

The temperature sensitivity coefficient at 9.95 m of the FUT was calculated to be 3.18 GHz/°C, as shown in Figure 6a. As shown by the olive line in Figure 5d, the spectral shift at 9.95 m was approximately 1680 GHz when the temperature was 600 °C, corresponding to the temperature sensitivity coefficient. Besides, the calculated R^2^ was between 0.985 and 0.999 at the position of 9.7 to 10.0 m of the FUT, indicating that the demodulated temperature was almost equal to the applied temperature. Obviously, the fluctuation of R^2^ outside of the furnace was significantly greater than that inside the furnace. 

Furthermore, the spectral shift variation, i.e., cumulative error along the FUT, of DM-1D, DM-2D, SM-1D and SM-2D, was also investigated when no temperature was applied. As shown in Figure 7, the corresponding mean and standard deviations calculated by DM-1D, DM-2D, SM-1D and SM-2D were 2.49 ± 0.70, 2.50 ± 0.36, 4.77 ± 1.51 and 4.51 ± 0.91 GHz, as represented by the pink, red, orange and green curves, respectively. This indicated that the SM could effectively expand the temperature demodulation range, i.e., a larger measurement range, but the cumulative error was larger than that of DM. The spectral shift caused by the cumulative error for SM-2D was 4.51 GHz, and the accuracy of the demodulated temperature was 0.91 GHz, corresponding to temperatures of 1.18 and 0.24 °C. 

## 4. Conclusions

In conclusion, a SM combined with IWDM, i.e., 2D method, was proposed and demonstrated to obtain a high-spatial-resolution OFDR distributed temperature sensor. Compared with DM-1D, DM-2D, SM-1D, the temperature distribution could be successfully demodulated by using DM-2D at a spatial resolution of 3.2 mm, when the temperature was increased from 50 to 600 °C. The temperature sensitivity coefficient of the Au-SMF was 3.18 GHz/°C. The spectral shift caused by the cumulative error was 1.18 °C and the accuracy of the demodulated temperature using SM-2D was 0.24 °C. Such a method has great potential to expand the temperature measurement range, which is very useful for high-temperature applications.

## Figures and Tables

**Figure 1 sensors-22-09972-f001:**
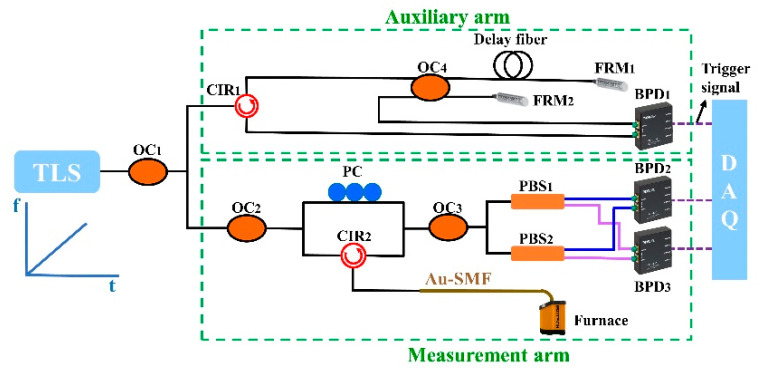
Optical frequency domain reflectometry (OFDR) experimental setup for distributed temperature sensing (DTS) using an Au-coated single mode fiber (Au-SMF), consisting of an auxiliary arm and measurement arm. TLS: tunable laser source; OC: optical fiber coupler; CIR: circulator; FRM: Faraday rotating mirror; BPD: balanced photodetector; PC: polarization controller; PBS: polarization beam splitter; Au-SMF: Au-coated single mode fiber; DAQ: data acquisition.

**Figure 2 sensors-22-09972-f002:**
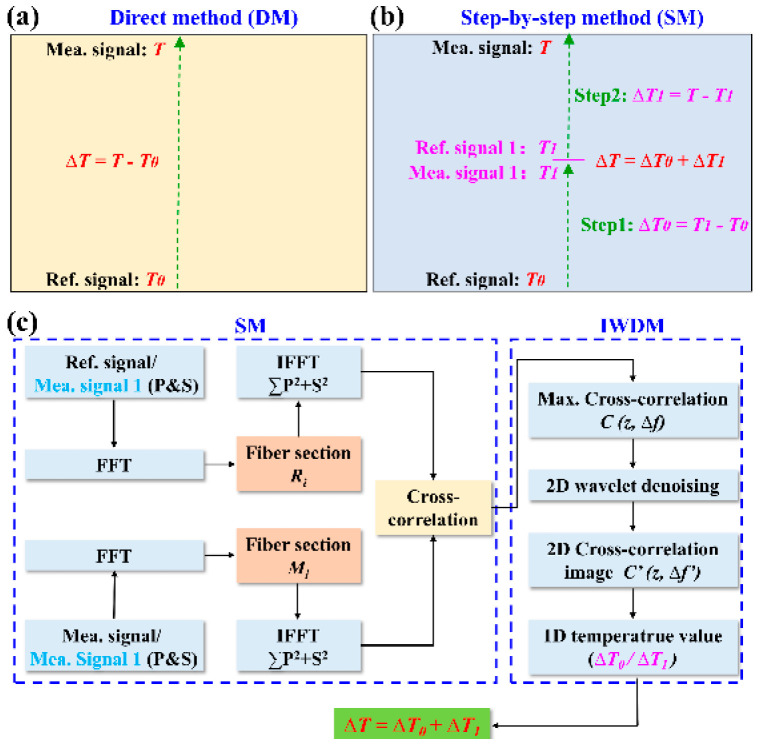
Schematic diagram of (**a**) direct method (DM) and (**b**) step-by-step method (SM); (**c**) flow diagram of temperature data processing by use of the SM and image wavelet denoising method (IWDM). Ref. signal: reference signal; Mea. signal: measurement signal; FFT: fast Fourier transformation; IFFT: inverse FFT; 2D: two-dimensional; 1D: one-dimensional.

**Figure 3 sensors-22-09972-f003:**
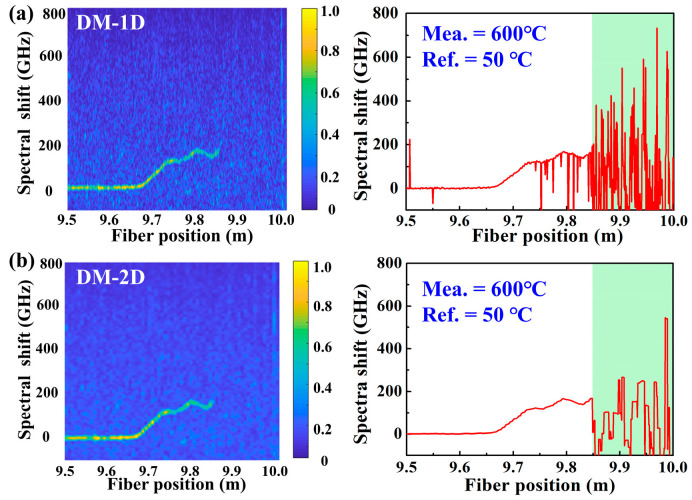
Temperature distribution demodulated by using the DM combined with 1D and 2D methods, i.e., (**a**) DM-1D, (**b**) DM-2D, respectively, at the temperature of 600 °C. The FUT between 9.85 and 10.0 m was placed in the furnace, represented by the cyan box. Note that in the DM, the Ref. signal and Mea. signal were defined as signals acquired at the temperature of 50 °C and 600 °C, respectively.

**Figure 4 sensors-22-09972-f004:**
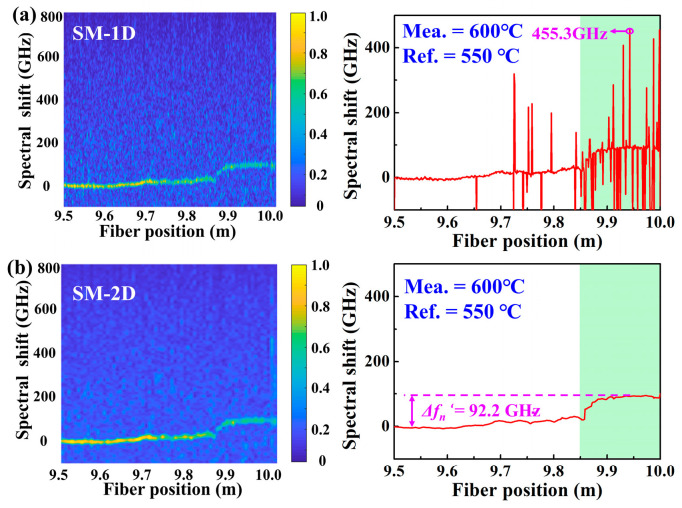
Temperature distribution demodulated by using the SM combined with 1D and 2D method, i.e., (**a**) SM-1D, (**b**) SM-2D, respectively, at the temperature of 600 °C. Note that, in the SM, the Ref. signal and Mea. signal was defined as signals acquired at the temperatures of 550 °C and 600 °C, respectively.

**Figure 5 sensors-22-09972-f005:**
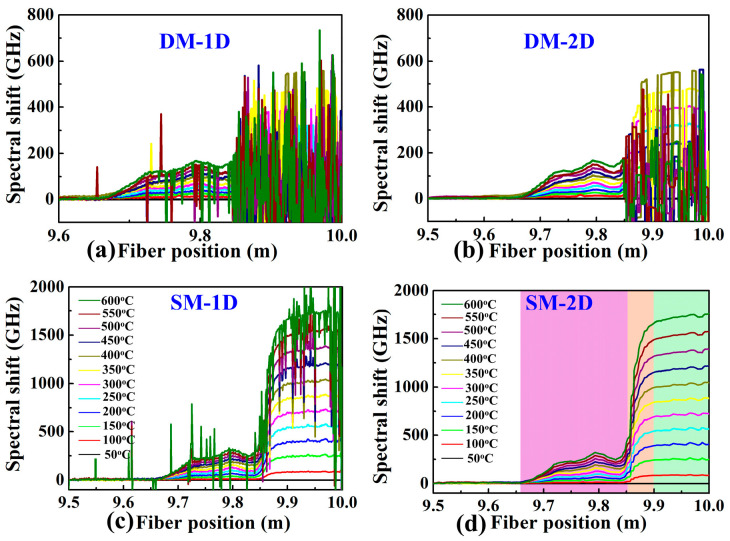
Spectral shift calculated by (**a**) DM-1D, (**b**) DM-2D, (**c**) SM-1D and (**d**) SM-2D at a spatial resolution of 3.2 mm, when the temperature was increased from 50 to 600 °C with a step of 50 °C. Note that the spatial resolution was 3.2 mm. The outside, edge and inside of the furnace are represented by the pink, orange and cyan boxes, respectively.

**Figure 6 sensors-22-09972-f006:**
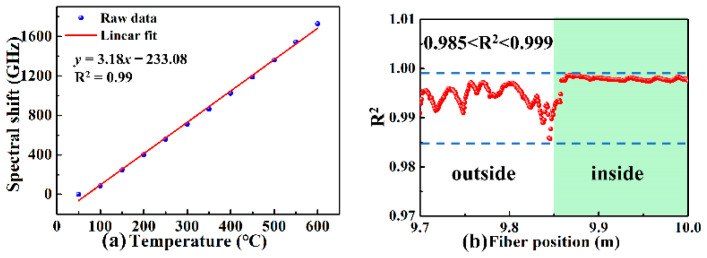
(**a**) Measured spectral shift as a function of the applied temperature at 9.95 m of the FUT and (**b**) 9.7–10 m when the furnace temperature increased from 50 to 600 °C.

**Figure 7 sensors-22-09972-f007:**
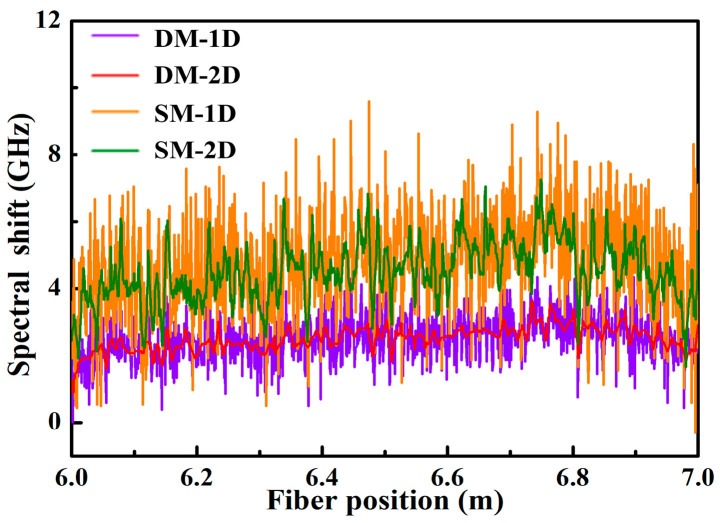
The variation of spectral shift obtained from the data processed by the DM-1D, DM-2D, SM-1D and SM-2D methods, represented by the pink, red, orange and green curves, respectively, when no temperature was applied.

## Data Availability

Not applicable.

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
