# Peer review of "High-Spatial-Resolution OFDR Distributed Temperature Sensor Based on Step-by-Step and Image Wavelet Denoising Methods"

_sensors, 2022, doi:10.3390/s22249972_

Round 1

Reviewer 1 Report

Major Comments:

Fu et al. present a method they refer to as a step-by-step method for improving OFDR data reconstruction, followed by a technique to denoise the resultant 2D data visualization. The step-by-step method has been already published on extensively in literature by Sweeney et al. to resolve data at high temperatures using SMF-28 at under high strains [1,4], under neutron/gamma irradiation [1], and at temperatures to >1000°C, with and without FBGs [1,2,3] (agreeing with the improved performance shown in Fig. 5). This technique was also first presented in a more general sense in [1] and formalized to presented as a network traversal problem and develop a provably optimal solution to the problem of OFDR reconstruction in a manuscript published in this same journal [2]. This context is not mentioned or appropriately discussed in the introduction and discussion. Such a discussion and presentation are critical to the work the authors present because this technique has been extensively developed elsewhere and is not novel to this work.

Other than this, the manuscript is concise and well-written and the abstraction of the frequency shift vs position 2D analysis is novel and important. The temperature sensitivity coefficient

Minor Comments:

The “Experimental setup and Methods” section, OFDR relies on extensive signal processing to recover a spectral shift measurement, and much of this may not be obvious to the average reader. I suggest developing the mathematical/physical framework using a more detailed description (including equations) to provide a more robust explanation of the procedure.  

The authors should provide a diagram of their experimental setup with a description of how the temperatures were measured to determine the temperature sensitivity coefficient out to three significant figures. Generally, furnaces do not have a uniform heating profile and can vary wildly along their lengths, in addition to natural circulation/buffeting issues during testing. Furthermore, a description of how the logged temperature corresponded to the fiber spectral shift is necessary to improve how this data is interpreted.

The wavelet denoising method the authors demonstrate is certainly interesting to remove spurious jumps within the data. However, because the authors are suggesting using a cumulative step-by-step/adaptive method to reconstruct their spectral shift data, the impact of the wavelet denoising method they present is unclear and should be developed further. This could be a very important result if the authors can demonstrate how denoising methods affect error propagation and/or signal drift from cumulative errors.

References:

[1] Sweeney, D. C., Schrell, A. M., & Petrie, C. M. (2020). An adaptive reference scheme to extend the functional range of optical backscatter reflectometry in extreme environments. IEEE Sensors Journal21(1), 498-509.

[2] Sweeney, D. C., Sweeney, D. M., & Petrie, C. M. (2021). Graphical optimization of spectral shift reconstructions for optical backscatter reflectometry. Sensors21(18), 6154.

[3] Jones, J. T., Sweeney, D. C., Birri, A., Petrie, C. M., & Blue, T. E. (2022). Calibration of distributed temperature sensors using commercially available SMF-28 optical fiber from 22° C to 1000° C. IEEE Sensors Journal22(5), 4144-4151.

[4] Sweeney, D. C., & Petrie, C. M. (2022). Expanding the range of the resolvable strain from distributed fiber optic sensors using a local adaptive reference approach. Optics Letters47(2), 269-272.

Author Response

1.1 Comment:

Fu et al. present a method they refer to as a step-by-step method for improving OFDR data reconstruction, followed by a technique to denoise the resultant 2D data visualization. The step-by-step method has been already published on extensively in literature by Sweeney et al. to resolve data at high temperatures using SMF-28 at under high strains [1,4], under neutron/gamma irradiation [1], and at temperatures to >1000°C, with and without FBGs [1,2,3] (agreeing with the improved performance shown in Fig. 5). This technique was also first presented in a more general sense in [1] and formalized to presented as a network traversal problem and develop a provably optimal solution to the problem of OFDR reconstruction in a manuscript published in this same journal [2]. This context is not mentioned or appropriately discussed in the introduction and discussion. Such a discussion and presentation are critical to the work, the authors present because this technique has been extensively developed elsewhere and is not novel to this work. Other than this, the manuscript is concise and well-written and the abstraction of the frequency shift vs position 2D analysis is novel and important. The temperature sensitivity coefficient.

Response:

Many thanks for your comments. The presentation on the afore-mentioned references have been added in the introduction.

[1] Sweeney, D. C., Schrell, A. M., & Petrie, C. M. (2020). An adaptive reference scheme to extend the functional range of optical backscatter reflectometry in extreme environments. IEEE Sensors Journal, 21(1), 498-509.

[2] Sweeney, D. C., Sweeney, D. M., & Petrie, C. M. (2021). Graphical optimization of spectral shift reconstructions for optical backscatter reflectometry. Sensors, 21(18), 6154.

[3] Jones, J. T., Sweeney, D. C., Birri, A., Petrie, C. M., & Blue, T. E. (2022). Calibration of distributed temperature sensors using commercially available SMF-28 optical fiber from 22° C to 1000° C. IEEE Sensors Journal, 22(5), 4144-4151.

[4] Sweeney, D. C., & Petrie, C. M. (2022). Expanding the range of the resolvable strain from distributed fiber optic sensors using a local adaptive reference approach. Optics Letters, 47(2), 269-272.

1.2 Comment:

The “Experimental setup and Methods” section, OFDR relies on extensive signal processing to recover a spectral shift measurement, and much of this may not be obvious to the average reader. I suggest developing the mathematical/physical framework using a more detailed description (including equations) to provide a more robust explanation of the procedure.

Response:

Many thanks for your comments. A detailed description on the principle of the OFDR has been added in the revised manuscript.

The signal from the AI acquired by the DAQ could be given by [10]

    (1)

where  is the delay time between the two arms of the auxiliary arm,  is the reflectivity with the fiber attenuation at the delay time of ,  and  are the initial optical frequency and sweep rate of the tunable laser source (TLS),  is the amplitude of the optical electric field,  is the phase noise term, respectively. In addition, the value of the beat frequency, i.e., , is proportional to the time delay, i.e., , with a relationship of . By the cross-correlation between the measurement Rayleigh backscattering (RBS) and the reference RBS, the RBS shifts, i.e., , could be obtained. Then,  caused by the temperature variation, i.e.,  could be given by

Therefore, the temperature variation along the FUT could be obtained by measuring the spectral shift, i.e.,

1.3 Comment:

The authors should provide a diagram of their experimental setup with a description of how the temperatures were measured to determine the temperature sensitivity coefficient out to three significant figures. Generally, furnaces do not have a uniform heating profile and can vary wildly along their lengths, in addition to natural circulation/buffeting issues during testing. Furthermore, a description of how the logged temperature corresponded to the fiber spectral shift is necessary to improve how this data is interpreted.

Response:

Many thanks for your comments. The temperature sensing property of the Au-SMF was investigated by placing the fiber end, i.e., a length of 10 cm, in the furnace (Gemini4857A). The furnace used in the experiment was illustrated in Fig. 1. The temperature was increased from 50 to 600℃ with a step of 50℃, remaining for 40 min at each temperature measurement point. Moreover, the diameter of the furnace cavity is 8 mm and only the top has an opening end, which makes the temperature in the furnace relatively stable. The accuracy of the furnace was ±0.05°C. As shown in Fig. 5(d) in the manuscript, an obvious non-uniform temperature profile between 9.65 and 9.90 m, i.e., outside and top of the furnace cavity, was obtained, while a relatively uniform temperature profile between 9.90 and 10.0 m, i.e., inside the furnace cavity, was obtained. This indicated that the range of non-uniform and uniform temperature profile was the same when heated from 50 to 600℃, i.e., 9.65-9.90 m and 9.90-10.0 m. In addition, the spectral shift at 9.95 m was approximately 1680.3 GHz, when the temperature was 600 °C, corresponding to the temperature sensitivity coefficient.

Fig. 1 The furnace used in the experiment

1.4 Comment:

The wavelet denoising method the authors demonstrate is certainly interesting to remove spurious jumps within the data. However, because the authors are suggesting using a cumulative step-by-step/adaptive method to reconstruct their spectral shift data, the impact of the wavelet denoising method they present is unclear and should be developed further. This could be a very important result if the authors can demonstrate how denoising methods affect error propagation and/or signal drift from cumulative errors.

Response:

The detailed wavelet denoising method was listed as follows. A two-dimensional (2D) cross-correlation matrix, i.e., C(z, ∆f), was decomposed into sub-images containing multiple frequency bands using discrete wavelet changes. Meantime, the noise signal could be eliminated by adopting a threshold function due to different wavelet coefficients of the noise and sensing signal. In other words, the wavelet coefficients below the threshold are the noise signal and set to zero, while above the threshold was the sensing signal. In this way, an optimized two-dimensional (2D) cross-correlation image matrix, i.e., C’(z, ∆f’), could be obtained by using the remained wavelet coefficients through inverse wavelet transform. Note that the threshold was set to 0.0664, i.e., three times of the obtained noise standard deviation, i.e., 0.0197. As shown in Fig. 5, the fluctuation of the spectral shift induced by temperature change could be eliminated. Thus, the spectral shift could be well distinguished.

Reviewer 2 Report

Dear authors, your article tackles an important limitation encountered with OFDR monitoring of elevated temperatures. However, it requires many corrections.

First, the title may be simplified "... based on step-by-step and image wavelet denoising methods". Electromagnetic immunity is not the only advantage provided by optical sensing, you should cite distributed monitoring as well.

Please pay attention to the tense. You should use the present tense anytime the action is still true today of if it is a timeless truth. For example page 1 - line 45 : "the DTS using FBG arrays IS still quasi-distributed, ...".

"... to improve the spatial resolution of the DTS based on OFDR."

"The system consisted in two arms : ..."

What is the purpose of OC3 ? Usually only one PBS is used to separate p and s components of the interferogram. Why using 2 PBS ?

The cross-correlation seems been done in the frequency domain, so mathematically speaking it is not a correlation. Please elaborate a bit more.

In Fig. 3, it is no surprise that the correlation fails with a so huge temperature shift (600 °C - 50°C). Since you chose two different reference T° it is not possible to distinguish the action of the algorithm itself. You should have used the same reference T° for both algorithms to provide a sound comparison.

Author Response

2.1 Comment:

Dear authors, your article tackles an important limitation encountered with OFDR monitoring of elevated temperatures. However, it requires many corrections. First, the title may be simplified "... based on step-by-step and image wavelet denoising methods". Electromagnetic immunity is not the only advantage provided by optical sensing, you should cite distributed monitoring as well.

Response:

Many thanks for your suggestions. The title has been simplified to ‘High-spatial-resolution OFDR distributed temperature sensor based on step-by-step and image wavelet denoising methods’. The advantages of distributed measurement capability, long sensing range and electromagnetic interference immunity have been added. (See lines 27-28)

2.2 Comment:

Please pay attention to the tense. You should use the present tense anytime the action is still true today of if it is a timeless truth. For example page 1 - line 45 : "the DTS using FBG arrays IS still quasi-distributed, ..."."... to improve the spatial resolution of the DTS based on OFDR." "The system consisted in two arms : ..."

Response:

Many thanks for your suggestions. The tense describing a timeless truth has been revised from past tense to present tense in the manuscript.

2.3 Comment:

What is the purpose of OC3? Usually only one PBS is used to separate p and s components of the interferogram. Why using 2 PBS ?

Response:

Many thanks for your comments. The optical coupler, i.e., OC3, was used to combine the light transmitted by polarization controller (PC)and light reflected by Au-SMF and. Then the two polarization beam splitters (PBSs), i.e., PBS1 and PBS2, was used to split the acquired signal into p/s polarization light. Moreover, the PBSs were also used to mitigate the polarization signal fading of the measurement arm [1].

[1] Ding, Zhenyang, Jiang, et al. Distributed Strain and Temperature Discrimination Using Two Types of Fiber in OFDR[J]. IEEE Photonics Journal, 2016.

2.4 Comment:

The cross-correlation seems been done in the frequency domain, so mathematically speaking it is not a correlation. Please elaborate a bit more.

Response:

Many thanks for your comments. In the experiment, the Ref. signal is the beat frequency interference signal acquired when the temperature is T1. Then the Mea. signal is the beat frequency interference signal acquired when the temperature is increased to T2. The cross-correlation is performed in the frequency domain of two signals, i.e., Ref. signal and Mea. signal.

2.5 Comment:

In Fig. 3, it is no surprise that the correlation fails with a so huge temperature shift (600 °C - 50°C). Since you chose two different reference T° it is not possible to distinguish the action of the algorithm itself. You should have used the same reference T° for both algorithms to provide a sound comparison.

Response:

Many thanks for your comments. In Fig. 3(a) and 3(b), the Ref. and Mea. signal is the signal acquired at the temperature of 50°C and 600°C, respectively. Thus, the same reference and measurement temperature is selected. The difference is that one is to use direct method (DM) combined with 1D method, i.e., DM-1D, and the other is DM combined with 2D method, i.e., DM-2D.

Round 2

Reviewer 2 Report

Dear authors, you provided many improvements to the text (please reconsider the paragraph in page 2 from line 49 to 58 that still contains mistakes (senor-to-sensor, However, ...). The additional text dedicated to OFDR analysis in page 3 is welcome but, once again, you should have cited at least one original paper from Luna's team (Froggatt, Soller etc.).